# Knowledge Distillation Based on Transformed Teacher Matching

**Kaixiang Zheng & En-Hui Yang**
Department of Electrical and Computer Engineering, University of Waterloo
{k56zheng,ehyang}@uwaterloo.ca

## Abstract

As a technique to bridge logit matching and probability distribution matching, temperature scaling plays a pivotal role in knowledge distillation (KD). Conventionally, temperature scaling is applied to both teacher's logits and student's logits in KD. Motivated by some recent works, in this paper, we drop instead temperature scaling on the student side, and systematically study the resulting variant of KD, dubbed transformed teacher matching (TTM). By reinterpreting temperature scaling as a power transform of probability distribution, we show that in comparison with the original KD, TTM has an inherent Rényi entropy term in its objective function, which serves as an extra regularization term. Extensive experiment results demonstrate that thanks to this inherent regularization, TTM leads to trained students with better generalization than the original KD. To further enhance student's capability to match teacher's power transformed probability distribution, we introduce a sample-adaptive weighting coefficient into TTM, yielding a novel distillation approach dubbed weighted TTM (WTTM). It is shown, by comprehensive experiments, that although WTTM is simple, it is effective, improves upon TTM, and achieves state-of-the-art accuracy performance. Our source code is available at https://github.com/zkxufo/TTM.

## 1 Introduction

Knowledge distillation (KD) has achieved a great success and drawn a lot of attention ever since it was proposed. The original form of KD was proposed by Buciluǎ et al. (2006), where a small model (student) was trained to match the logits of a large model (teacher). Later, a generalized version now known as KD was proposed by Hinton et al. (2015), where the small student model was trained to match the class probability distribution of the large teacher model. Compared to the student model trained with standard empirical risk minimization (ERM), the student model trained via KD has better performance in terms of accuracy, to the extent that this light-weight KD-trained student model is able to take the place of some larger and more complex models with little performance degradation, achieving the goal of model compression.

In the literature, KD is generally formulated as minimizing the following loss

$$\mathcal{L}_{KD} = (1 - \lambda)H(y, q) + \lambda T^2 D(p_T^t || q_T) \tag{1}$$

where $\mathcal{L}_{CE} = H(y, q)$ is the cross entropy loss between the one-hot probability distribution corresponding to label $y$ and the student output probability distribution $q$, which is the canonical loss of ERM, $D(p_T^t || q_T)$ is the Kullback–Leibler divergence between the temperature scaled output probability distribution $p_T^t$ of the teacher and the temperature scaled output probability distribution $q_T$ of the student, $T$ is the temperature of distillation, and $\lambda$ is a balancing weight. Note that $p_T^t = \sigma(v/T)$ and $q_T = \sigma(z/T)$, given logits $v$ of the teacher and logits $z$ of the student, where $\sigma$ denotes the softmax function.

The use of the temperature $T$ above is a pivotal characteristic of KD. On one hand, it provides a way to build a bridge between class probability distribution matching and logits matching. Indeed, it was shown in Hinton et al. (2015) that as $T$ goes to $\infty$, KD is equivalent to its logits-matching predecessor. On the other hand, it also distinguishes KD from the logits-matching approach, since in practice, empirically optimal values of the temperature $T$ are often quite modest. Beyond these, there

is little understanding about the role of the temperature $T$ and in general why KD in its formulation (1) helps the student learns better. In particular, the following questions naturally arise:

**Q1** Why does the temperature $T$ have to be applied to both the teacher and student?

**Q2** Would it be better off to apply the temperature $T$ to the teacher only, but not to the student?

So far, answers to the above questions remain elusive at the best.

The purpose of this paper is to address the above questions. First, we demonstrate both theoretically and experimentally that the answer to the question Q2 above is affirmative, and it is better off to drop the temperature $T$ entirely on the student side—the resulting variant of KD is referred to as transformed teacher matching (TTM) and formulated as minimizing the following objective:

$$\mathcal{L}_{TTM} = H(y, q) + \beta D(p_T^t || q) \tag{2}$$

where $\beta$ is a balancing weight. Specifically, we show that (1) temperature scaling of logits is equivalent to a power transform of probability distribution, and (2) in comparison with KD, TTM has an inherent Rényi entropy term in its objective function (2). It is this inherent Rényi entropy that serves as an extra regularization term and hence improves upon KD. This theoretic analysis is further confirmed by extensive experiment results. It is shown by extensive experiments that thanks to this inherent regularization, TTM leads to trained students with better generalization than KD. Second, to further enhance student's capability to match teacher's power transformed probability distribution, we introduce a sample-adaptive weighting coefficient into TTM, yielding a novel distillation approach dubbed weighted TTM (WTTM). WTTM is simple and has almost the same computational complexity as KD. And yet it is very effective; it is shown, by comprehensive experiments, that it is significantly better than KD in terms of accuracy, improves upon TTM, and achieves state-of-the-art accuracy performance. For example, WTTM can reach 72.19% classification accuracy on ImageNet for ResNet-18 distilled from ResNet-34, outperforming most highly complex feature-based distillation methods.

With the temperature $T$ dropped entirely on the student side, TTM and WTTM, along with the statistical perspective of KD (Menon et al., 2021) and the newly established upper bound on error rate in term of the cross entropy $H(p_x^*, q)$ between the true, but often unknown conditional probability distribution $p_x^*$ of label $y$ given an input sample $x$ and the output probability distribution $q$ of a model in response to the input $x$, Yang et al. (2023a) offer a new explanation of why KD helps. First, the purpose of the teacher in KD is to provide a proper estimate for the unknown true conditional probability distribution $p_x^*$, which is a linear combination of the one-hot vector corresponding to the label $y$ and the power transformed teacher's probability distribution $p_T^t$. Second, the role of the temperature $T$ on the teacher side is to improve this estimate. Third, replacing $p_x^*$ by its estimate from the transformed teacher, the learning process in KD is to simply minimize the cross entropy upper bound on error rate, which improves upon the standard deep learning process where $p_x^*$ in the cross entropy upper bound is rudimentarily approximated by the one-hot vector corresponding to the label $y$.

## 2 BACKGROUND AND RELATED WORK

### 2.1 CONFIDENCE PENALTY

In a multi-class classification setting, an output of a neural network in response to an input sample is a probability vector or distribution $q$ with $K$ entries, where $K$ is the number of all possible classes, and the class with the highest probability is the prediction made by the neural network for this particular sample. Conventionally, a prediction is said to be confident if the corresponding $q$ concentrates most of its probability mass on the predicted class. Szegedy et al. (2016) points out that if a model is too confident about its predictions, then it tends to suffer from overfitting. To avoid overfitting and improve generalization, Pereyra et al. (2017) proposed to penalize confident predictions. Since a confident prediction generally corresponds to $q$ with low entropy, they enforced confidence penalty (CP) by introducing a negative entropy regularizer into the objective function of the learning process, which is formulated as

$$\mathcal{L}_{CP} = H(y, q) - \eta H(q) \tag{3}$$

where $\eta$ controls the strength of the confidence penalty. Thanks to the entropy regularization, the learned model is encouraged to output smoother distributions with larger entropy, leading to less confident predictions, and most importantly, better generalization.

## 2.2 Rényi Entropy

Rényi entropy (Rényi, 1961) is a generalized version of Shannon entropy, which has been successfully applied in many machine learning topics, such as differential privacy (Mironov, 2017), understanding neural networks (Yu et al., 2020), and representation distillation (Miles et al., 2021). Given a discrete random variable $X$ with alphabet $\mathcal{A} = \{x_1, x_2, \dots, x_n\}$ and corresponding probabilities $p_i$ for $i = 1, 2, \dots, n$, its Rényi entropy is defined as

$$H_\alpha(X) = \frac{1}{1 - \alpha} \log \sum_{i=1}^{n} p_i{}^\alpha \tag{4}$$

where $\alpha$ is called the order of Rényi entropy. The limit of Rényi entropy when $\alpha \to 1$ is the well-known Shannon entropy.

## 2.3 Label Smoothing Perspective towards KD

In the literature, different perspectives have been developed to understand KD. One of them is the label smoothing (LS) perspective advocated by Yuan et al. (2020) and Zhang & Sabuncu (2020).

LS (Szegedy et al., 2016) is a technique to encourage a model to make less confident predictions by minimizing the following objective function in the learning process

$$\mathcal{L}_{LS} = (1 - \epsilon)H(y, q) + \epsilon H(u, q) \tag{5}$$

where $u$ is a uniform distribution over all $K$ possible classes, and $\epsilon$ controls the strength of the smoothing effect. The model trained with LS tends to have significantly less confident predictions and output probability distributions with larger Shannon entropy compared to its counterpart in the case of ERM (visualized in A.1).

If we replace $u$ with the teacher output $p^t$ in (5), then we have $\mathcal{L}_{LS} = (1 - \epsilon)H(y, q) + \epsilon H(p^t, q)$, which is equivalent to $\mathcal{L}_{KD}$ with $T = 1$, since the entropy $H(p^t)$ does not depends on the student. Therefore, when $T = 1$, KD can indeed be regarded as sample-adaptive LS. However, when $T > 1$, such a perspective no longer holds since temperature scaling is also applied to the student model. This is confirmed by the empirical analysis shown in A.1. Although KD with $T = 1$ is able to increase the Shannon entropy of output probability distribution $q$ compared to ERM, KD with $T = 4$ actually leads to decreased Shannon entropy compared to ERM, showing an opposite effect of LS.

The sample-adaptive LS perspective was also advocated in self-distillation Zhang & Sabuncu (2020), where the temperature $T$ was dropped for convenience on the student side. However, no systematic treatment was provided to justify the drop-out of the temperature $T$ for the student side. In fact, in terms of prediction accuracy, mixed results were demonstrated: dropping out the temperature $T$ for the student can either decrease or increase the accuracy.

## 2.4 Statistical Perspective and Cross Entropy Upper Bound

Another perspective to understand KD is the statistical perspective advocated by Menon et al. (2021). A key observation therein is that the Bayes-distilled risk has a smaller variance than the standard empirical risk, which is actually the direct consequence of the law of total probability for variance (Ross, 2019). Since the Bayes class-probability distribution over the labels, i.e., the conditional probability distribution $p_x^* = [P(i|x)]_{i=1}^{K}$ of label $y$ given an input sample $x$, is unknown in practice, the role of the teacher in KD was believed to use its output probability distribution $p^t$ or temperature scaled output probability distribution $p_T^t$ to estimate $p_x^*$ for the student. This, in turn, offers some explanation of why improving teacher accuracy can sometimes harm distillation performance, since improving teacher accuracy and providing better estimates for $p_x^*$ are two different tasks. In this perspective, the temperature $T$ is also dropped for the student. Again, no justification was provided for dropping $T$ on the student side. In addition, the question of why minimizing the Bayes-distilled risk or teacher-distilled risk could improve the student's accuracy performance was not answered either.

Recently, it was shown in Yang et al. (2023a) that for any classification neural network, its error rate is upper bounded by $\mathbb{E}_x[H(p_x^*, q)]$. Thus, to reduce its error rate, the neural network can be trained by minimizing $\mathbb{E}_x[H(p_x^*, q)]$. Since the true conditional distribution $p_x^*$ is generally unavailable in practice, KD with the temperature $T$ dropped for the student can be essentially regarded as one way

to solve approximately the problem of minimizing $\mathbb{E}_x[H(p_x^*, q)]$, where $p_x^*$ is first approximated by a linear combination of the one-hot probability distribution corresponding to label $y$ and the temperature scaled output probability distribution $p_T^t$ of the teacher. This perspective, when applied to KD, does provide justifications for dropping the temperature $T$ entirely on the student side and also for minimizing the Bayes-distilled risk or teacher-distilled risk. Of course, KD with the temperature $T$ dropped for the student may not be necessarily an effective way to minimize $\mathbb{E}_x[H(p_x^*, q)]$. Other recent related works are reviewed in Appendix A.7.

In contrast, in this paper, we show more directly that it is better off to drop entirely the temperature $T$ on the student side in KD by comparing TTM with KD both theoretically and experimentally.

## 3 TRANSFORMED TEACHER MATCHING

In this section, we compare TTM with KD theoretically by showing that TTM is equivalent to KD plus Rényi entropy regularization. To this end, we first come up with a general concept of power transform of output distributions. Then, we show the equivalence between temperature scaling and power transform. Based on this, a simple derivation is provided to decompose TTM into KD plus a Rényi entropy regularizer. In view of CP, it's clear that TTM can lead to better generalization than KD because of the penalty over confident output distributions.

### 3.1 POWER TRANSFORM OF PROBABILITY DISTRIBUTIONS

In KD, model output distributions are transformed by temperature scaling to improve their smoothness. However, such a transform is not unique. There are many other transforms which can smooth out peaked probability distributions as well. Below we will introduce a generalized transform.

Consider a point-wise mapping $f : [0, 1] \to [0, 1]$. For any probability distribution $p = [p_1, \ldots, p_K]$, we can apply $f$ to each component of $p$ to define a generalized transform $p \to \hat{p}$, where $\hat{p} = [\hat{p_1}, \ldots, \hat{p_K}]$, and

$$\hat{p_i} = \frac{f(p_i)}{\sum_{j=1}^{K} f(p_j)}, \; \forall\, 1 \le i \le K. \tag{6}$$

In this above, $\sum_{j=1}^{K} f(p_j)$ is used to normalize the vector $[f(p_i)]_{i=1}^{K}$ back to a probability simplex. With this generalized framework, any specific transform can be described by its associated mapping $f$. Among all possible mappings $f$, the most interesting one to us is the power function with exponent $\gamma$. If $f$ is selected to be the power function with exponent $\gamma$, the resulting probability distribution transform $p \to \hat{p}$ is referred to as the power transform of probability distribution. Accordingly, the power transformed distribution is given by

$$\hat{p} = [\hat{p_i}]_{i=1}^{K} = \left[ \frac{p_i{}^{\gamma}}{\sum_{j=1}^{K} p_j{}^{\gamma}} \right]_{i=1}^{K}. \tag{7}$$

Next, we will show that power transform is equivalent to temperature scaling. Indeed, suppose that $p$ is the softmax of logits $[l_1, l_2, \cdots, l_K]$:

$$p_i = \frac{e^{l_i}}{\sum_{j=1}^{K} e^{l_j}}, \; \forall\, 1 \le i \le K. \tag{8}$$

Then

$$\hat{p_i} = \frac{p_i{}^{\gamma}}{\sum_j p_j{}^{\gamma}} = \frac{\left( \frac{e^{l_i}}{\sum_m e^{l_m}} \right)^{\gamma}}{\sum_j \left( \frac{e^{l_j}}{\sum_k e^{l_k}} \right)^{\gamma}} = \frac{\left( \frac{1}{\sum_m e^{l_m}} \right)^{\gamma} \cdot e^{\gamma l_i}}{\left( \frac{1}{\sum_k e^{l_k}} \right)^{\gamma} \cdot \sum_j e^{\gamma l_j}} = \frac{e^{\gamma l_i}}{\sum_j e^{\gamma l_j}}. \tag{9}$$

Thus $\hat{p}$ is the softmax of the scaled logits $[\gamma l_1, \gamma l_2, \cdots, \gamma l_K]$ with temperature $T = 1/\gamma$.

### 3.2 FROM KD TO TTM

Based on the equivalence between power transform and temperature scaling, we can now reveal the connection between KD and TTM.

Let $\gamma = 1/T$. Go back to (1) and (2). In view of (9), we have

$$p_T^t = \hat{p}^t \text{ and } q_T = \hat{q}. \tag{10}$$

Then we can decompose $D(p_T^t || q_T)$ as follows:

$$D(p_T^t || q_T) = D(\hat{p}^t || \hat{q})$$

$$= \sum_i \hat{p}^t{}_i \log \frac{\hat{p}^t{}_i}{\hat{q}_i}$$

$$= -\sum_i \hat{p}^t{}_i \log \hat{q}_i - H(\hat{p}^t)$$

$$= -\sum_i \hat{p}^t{}_i \log \frac{q_i{}^\gamma}{\sum_j q_j{}^\gamma} - H(\hat{p}^t) \tag{11}$$

$$= -\sum_i \hat{p}^t{}_i \log q_i{}^\gamma + \log \sum_j q_j{}^\gamma - H(\hat{p}^t)$$

$$= \gamma H(\hat{p}^t, q) + (1-\gamma) H_\gamma(q) - H(\hat{p}^t) \tag{12}$$

$$= \gamma D(\hat{p}^t || q) + (1-\gamma) H_\gamma(q) - (1-\gamma) H(\hat{p}^t) \tag{13}$$

$$= \gamma D(p_T^t || q) + (1-\gamma) H_\gamma(q) - (1-\gamma) H(p_T^t) \tag{14}$$

where (11) follows the power transform (7), $H_\gamma(q)$ in (12) is the Rényi entropy of $q$ of order $\gamma$, and (14) is due to (10). Rearranging (14), we get

$$D(p_T^t || q) = T D(p_T^t || q_T) - (T-1) H_{\frac{1}{T}}(q) + (T-1) H(p_T^t). \tag{15}$$

Plugging (15) into (2) yields

$$\mathcal{L}_{TTM} = H(y, q) + \beta T D(p_T^t || q_T) - \beta(T-1) H_{\frac{1}{T}}(q) + \beta(T-1) H(p_T^t)$$

$$\equiv H(y, q) + \beta T D(p_T^t || q_T) - \beta(T-1) H_{\frac{1}{T}}(q) \tag{16}$$

$$= \frac{1}{1-\lambda} \left[ (1-\lambda) H(y, q) + \lambda T^2 D(p_T^t || q_T) - \lambda T(T-1) H_{\frac{1}{T}}(q) \right] \tag{17}$$

$$= \frac{1}{1-\lambda} \left[ \mathcal{L}_{KD} - \lambda T(T-1) H_{\frac{1}{T}}(q) \right] \tag{18}$$

whenever $\beta$ is selected to be

$$\beta = \frac{\lambda}{1-\lambda} T, \tag{19}$$

where (16) is due to the fact that the Shannon entropy $H(p_T^t)$ does not depend on the student model, (17) follows (19), and (18) is attributable to (1).

Thus we have shown that TTM can indeed be decomposed into KD plus a Rényi entropy regularizer. Since Rényi entropy is a generalized version of Shannon entropy, it plays a role in TTM similar to that of Shannon entropy in CP. With this, we have reasons to believe that it can lead to better generalization, which is indeed confirmed later by extensive experiments in Section 5.

It is also instructive to compare TTM and KD from the perspective of their respective gradients. The gradients of the distillation component in $\mathcal{L}_{TTM}$ with respect to the logits are:

$$\frac{\partial D(p_T^t || q)}{\partial z_i} = \frac{\partial H(p_T^t, q)}{\partial z_i} = q_i - \hat{p}^t{}_i = q_i - \frac{(p_i^t)^{1/T}}{\sum_{j=1}^K (p_j^t)^{1/T}} \tag{20}$$

where $z_i$ and $q_i$ are the $i$th logit and $i$th class probability of the student model, respectively. In comparison, the corresponding gradients for KD are

$$\frac{\partial D(p_T^t || q_T)}{\partial z_i} = \frac{\partial H(p_T^t, q_T)}{\partial z_i} = \frac{1}{T} \left( \hat{q}_i - \hat{p}^t{}_i \right) = \frac{1}{T} \left( \frac{q_i^{1/T}}{\sum_{j=1}^K q_j^{1/T}} - \frac{(p_i^t)^{1/T}}{\sum_{j=1}^K (p_j^t)^{1/T}} \right). \tag{21}$$

From Eq. (20), we see that the gradient descent learning process would push $q_i$ to move towards the power transformed teacher probability distribution, thus encouraging the student to behave like the power transformed teacher, from which the name TTM (transformed teacher matching) is coined. Since the power transformed teacher distribution $p_T^t$ with $T > 1$ is smoother, the student trained by TTM will output a distribution $q$ with similar smoothness, leading to low confidence and high entropy. On the other hand, in Eq. (21), it is the transformed student distribution $q_T$ that is pushed towards the transformed teacher distribution $p_T^t$. Even when $q_T$ has similar smoothness as $p_T^t$, the original student distribution $q$ can still be quite peaked, thus having high confidence and low entropy.

# 4 SAMPLE-ADAPTIVE MATCHING TO THE TRANSFORMED TEACHER

We can further improve TTM by introducing a sample-adaptive weighting coefficient into TTM. This is explored in this section.

In TTM, the soft target we use is a linear combination of the one-hot probability distribution corresponding to $y$ and the power transformed teacher distribution $p_T^t$, where the same coefficient $\beta$ is applied to all samples. As discussed in Subsection 2.4, the role of the teacher in KD is to provide $p_T^t$ and use it as an estimate for $p_x^*$. Assume this estimate is good. It is reasonable to believe that it would be better off to favor a soft target over an one-hot target even more for those samples for which $p_T^t$ have more intrinsic confusion and is away from the one-hot probability distribution. After all, when $p_T^t$ is close to the corresponding one-hot probability distribution, minimizing $H(p_T^t, q)$ has little difference from minimizing $H(y, q)$, and as a result, it's no longer meaningful to do distillation on these types of samples. This motivates us to discriminate among soft targets in TTM based on their smoothness. Concretely, a large $\beta$ should be assigned to a smooth $p_T^t$, while a small $\beta$ should be assigned to a peaked $p_T^t$.

To implement the above idea, we need a quantity to quantify the smoothness of a soft target $p_T^t$. In view of (7) and the definition of Rényi entropy (4), the following power sum defined for any distribution $p$ and any $0 < \gamma < 1$

$$U_\gamma(p) = \sum_{j=1}^{k} p_j^\gamma$$

comes handy. Given $0 < \gamma < 1$, we can use the power sum $U_\gamma(p)$ to quantify the smoothness of $p$, since it is related to both the power transform and Rényi entropy. It is clear that the power sum $U_\gamma(p)$ attains its minimum 1 when $p$ is one-hot and maximum $K^{1-\gamma}$ when $p$ is uniform. Using $U_\gamma(p^t)$ to discriminate among different samples, we modify TTM to minimize the following objective function

$$\mathcal{L}_{WTTM} = H(y, q) + \beta U_{\frac{1}{T}}(p^t) \cdot D(p_T^t || q). \tag{22}$$

The resulting variant of KD is referred to as weighted TTM (WTTM). Note that other sample-adaptive weights such as $H(p_T^t)$ may also be effective. Nonetheless, systematic study regarding how to select sample-adaptive weights and which one is optimal, is left for future work.

Compared to TTM where the student is trained to match all soft targets uniformly, WTTM trains the student to match more closely to smooth soft targets and less closely to peaked soft targets. Thus, students resulting from WTTM would output smoother $q$ than those distilled from TTM, which is further confirmed in the next section by experiments.

# 5 EXPERIMENTS

## 5.1 EXPERIMENTAL SETTINGS

We benchmark TTM and WTTM on two prevailing image classification datasets, namely CIFAR-100 and ImageNet (Deng et al., 2009).

**CIFAR-100** contains 60k 32×32 color images of 100 classes, with 600 images per class, and it's further split into 50k training images and 10k test images. For fair comparison, we adopt the same training strategy and teacher models as CRD (Tian et al., 2019). Also, following CRD, we generate comprehensive experiment results for 13 teacher-student pairs including both same-architecture distillation and different-architecture distillation, and the tested model architectures are VGG (Simonyan & Zisserman, 2014), ResNet (He et al., 2016), WideResNet (Zagoruyko & Komodakis, 2016b), MobileNetV2 (Sandler et al., 2018), and ShuffleNet (Zhang et al., 2018; Ma et al., 2018).

**ImageNet** is a large-scale image dataset consisting of over 1.2 million training images and 50k validation images from 1000 classes. For experiments on ImageNet, we employ torchdistill (Matsubara, 2021) library and follow all the standard settings. The tested model architectures are ResNet and MobileNet (Howard et al., 2017).

Note that we list $T$ and $\beta$ values of all experiments in A.4 for reproducibility.

## 5.2 MAIN RESULTS

**Results on CIFAR-100**. The pure performances of TTM and WTTM are shown in Table 1 and Table 3. We compare them with feature-based methods FitNet (Romero et al., 2014), AT (Zagoruyko & Komodakis, 2016a), VID (Ahn et al., 2019), RKD (Park et al., 2019), PKT (Passalis & Tefas, 2018), CRD (Tian et al., 2019), and logits-based methods such as KD, DIST (Huang et al., 2022) and DKD (Zhao et al., 2022). In general, TTM and WTTM provide outstanding performance among all the compared methods, and WTTM is better than TTM in most cases. Note that TTM always outperforms KD, confirming our theoretic analysis in Section 3.

To further improve the performance, we combine WTTM loss with 2 existing distillation losses respectively, namely CRD and ITRD (Miles et al., 2021), and the resulting performance is shown in Table 2 and Table 4. For the combined methods, we directly adopt the optimal hyperparameters specified in the original papers without tuning (see A.5 for details). From the tables, we can see that the performance of the combined loss is always better than the pure performances of both ingredient losses, meaning that our proposed WTTM loss is orthogonal to other losses like CRD and ITRD. More importantly, the performance of WTTM aided by CRD and ITRD is consistently better than all other methods over all teacher-student pairs, achieving the state-of-the-art accuracy.

Table 1: Top-1 accuracy (%) on CIFAR-100 of student models trained with various distillation methods, including both feature-based methods and logits-based methods. Each teacher-student pair has the **same** architecture. We highlight the best results in bold, and the second best results with underscores. Note that some results of DIST (for the models excluded in their paper) are produced by our reimplementation. Average over 5 runs.

| Teacher
Student | WRN-40-2
WRN-16-2 | WRN-40-2
WRN-40-1 | resnet56
resnet20 | resnet110
resnet20 | resnet110
resnet32 | resnet32x4
resnet8x4 | vgg13
vgg8 |
|---|---|---|---|---|---|---|---|
| Teacher | 75.61 | 75.61 | 72.34 | 74.31 | 74.31 | 79.42 | 74.64 |
| Student | 73.26 | 71.98 | 69.06 | 69.06 | 71.14 | 72.50 | 70.36 |
| *Feature-based* | | | | | | | |
| FitNet | 73.58 | 72.24 | 69.21 | 68.99 | 71.06 | 73.50 | 71.02 |
| AT | 74.08 | 72.77 | 70.55 | 70.22 | 72.31 | 73.44 | 71.43 |
| VID | 74.11 | 73.30 | 70.38 | 70.16 | 72.61 | 73.09 | 71.23 |
| RKD | 73.35 | 72.22 | 69.61 | 69.25 | 71.82 | 71.90 | 71.48 |
| PKT | 74.54 | 73.45 | 70.34 | 70.25 | 72.61 | 73.64 | 72.88 |
| CRD | 75.48 | 74.14 | 71.16 | 71.46 | 73.48 | 75.51 | 73.94 |
| *Logits-based* | | | | | | | |
| KD | 74.92 | 73.54 | 70.66 | 70.67 | 73.08 | 73.33 | 72.98 |
| DIST | 75.51 | 74.73 | 71.75 | 71.65 | 73.69 | 76.31 | 73.89 |
| DKD | 76.24 | **74.81** | **71.97** | n/a | 74.11 | **76.32** | **74.68** |
| **TTM** | 76.23 | 74.32 | 71.83 | 71.46 | 73.97 | 76.17 | 74.33 |
| **WTTM** | **76.37** | 74.58 | 71.92 | **71.67** | **74.13** | 76.06 | 74.44 |

Table 2: Top-1 accuracy (%) on CIFAR-100. Each teacher-student pair has the **same** architecture. Average over 5 runs (3 runs for ITRD and **WTTM**+ITRD following the original paper of ITRD).

| Teacher
Student | WRN-40-2
WRN-16-2 | WRN-40-2
WRN-40-1 | resnet56
resnet20 | resnet110
resnet20 | resnet110
resnet32 | resnet32x4
resnet8x4 | vgg13
vgg8 |
|---|---|---|---|---|---|---|---|
| CRD | 75.48 | 74.14 | 71.16 | 71.46 | 73.48 | 75.51 | 73.94 |
| ITRD | 76.12 | 75.18 | 71.47 | 71.99 | 74.26 | 76.19 | 74.93 |
| **WTTM** | 76.37 | 74.58 | 71.92 | 71.67 | 74.13 | 76.06 | 74.44 |
| **WTTM+CRD** | 76.61 | 74.94 | **72.20** | 72.13 | **74.52** | 76.65 | 74.71 |
| **WTTM+ITRD** | **76.65** | **75.34** | 72.16 | **72.20** | 74.36 | **77.36** | **75.13** |

**Results on ImageNet**. In Table 5, we demonstrate the performance of WTTM compared to many competitive distillation methods such as KD, CRD, SRRL (Yang et al., 2020), ReviewKD (Chen et al., 2021), ITRD (Miles et al., 2021), DKD (Zhao et al., 2022), DIST (Huang et al., 2022), KD++ (Wang et al., 2023), NKD (Yang et al., 2023b), CTKD (Li et al., 2023c), and KD-Zero (Li et al., 2023a). It's shown that WTTM achieves outstanding performance on both teacher-student pairs.

## 5.3 EXTENSIONS

To provide more comprehensive understanding and deeper insight about TTM and WTTM, we include 4 points of extension in this subsection, demonstrating some promising properties of WTTM and supporting our methodology with some analysis.

Table 3: Top-1 accuracy (%) on CIFAR-100. Each teacher-student pair has **different** architectures. Note that some results of DIST (for the models excluded in their paper) are produced by our reimplementation. Average over 3 runs.

| Teacher
Student | vgg13
MobileNetV2 | ResNet50
MobileNetV2 | ResNet50
vgg8 | resnet32x4
ShuffleNetV1 | resnet32x4
ShuffleNetV2 | WRN-40-2
ShuffleNetV1 |
|---|---|---|---|---|---|---|
| Teacher | 74.64 | 79.34 | 79.34 | 79.42 | 79.42 | 75.61 |
| Student | 64.6 | 64.6 | 70.36 | 70.5 | 71.82 | 70.5 |
| *Feature-based* | | | | | | |
| FitNet | 64.14 | 63.16 | 70.69 | 73.59 | 73.54 | 73.73 |
| AT | 59.40 | 58.58 | 71.84 | 71.73 | 72.73 | 73.32 |
| VID | 65.56 | 67.57 | 70.30 | 73.38 | 73.40 | 73.61 |
| RKD | 64.52 | 64.43 | 71.50 | 72.28 | 73.21 | 72.21 |
| PKT | 67.13 | 66.52 | 73.01 | 74.10 | 74.69 | 73.89 |
| CRD | **69.73** | 69.11 | 74.30 | 75.11 | 75.65 | 76.05 |
| *Logits-based* | | | | | | |
| KD | 67.37 | 67.35 | 73.81 | 74.07 | 74.45 | 74.83 |
| DIST | 68.50 | 68.66 | 74.11 | 76.34 | **77.35** | 76.40 |
| DKD | 69.71 | **70.35** | n/a | 76.45 | 77.07 | **76.70** |
| **TTM** | 68.98 | 69.24 | **74.87** | 74.18 | 76.57 | 75.39 |
| **WTTM** | 69.16 | 69.59 | 74.82 | 74.37 | 76.55 | 75.42 |

Table 4: Top-1 accuracy (%) on CIFAR-100. Each teacher-student pair has **different** architectures. Average over 3 runs.

| Teacher
Student | vgg13
MobileNetV2 | ResNet50
MobileNetV2 | ResNet50
vgg8 | resnet32x4
ShuffleNetV1 | resnet32x4
ShuffleNetV2 | WRN-40-2
ShuffleNetV1 |
|---|---|---|---|---|---|---|
| CRD | 69.73 | 69.11 | 74.30 | 75.11 | 75.65 | 76.05 |
| ITRD | 70.39 | 71.41 | 75.71 | 76.91 | 77.40 | 77.35 |
| **WTTM** | 69.16 | 69.59 | 74.82 | 74.37 | 76.55 | 75.42 |
| **WTTM+CRD** | 70.30 | 70.84 | 75.30 | 75.82 | 77.04 | 76.86 |
| **WTTM+ITRD** | **70.70** | **71.56** | **76.00** | **77.03** | **77.68** | **77.44** |

Table 5: Top-1 accuracy (%) on ImageNet. The adopted teacher models are released by PyTorch (Paszke et al., 2019).

| Teacher | Student | KD | CRD | SRRL | ReviewKD | ITRD | DKD | DIST | KD++ | NKD | CTKD | KD-Zero | **WTTM** |
|---|---|---|---|---|---|---|---|---|---|---|---|---|---|
| ResNet-34 (73.31) | ResNet-18 (69.76) | 70.66 | 71.17 | 71.73 | 71.61 | 71.68 | 71.70 | 72.07 | 71.98 | 71.96 | 71.51 | 72.17 | **72.19** |
| ResNet-50 (76.16) | MobileNet (68.87) | 70.50 | 71.37 | 72.49 | 72.56 | n/a | 72.05 | **73.24** | 72.77 | 72.58 | n/a | 73.02 | 73.09 |

**Distill without $\mathcal{L}_{CE}$.** In Table 6, we compare the performance of WTTM **without** $\mathcal{L}_{CE}$ to the performance of KD **with** $\mathcal{L}_{CE}$. We find that even in this unfair setting, WTTM can still outperform KD in most cases. This is of great value in the scenario where the ground-truth labels of the transfer set are not available.

Table 6: Comparison between WTTM **without** $\mathcal{L}_{CE}$ and KD **with** $\mathcal{L}_{CE}$ on CIFAR-100. Accuracy is averaged over 5 runs.

| Teacher
Student | WRN-40-2
WRN-16-2 | WRN-40-2
WRN-40-1 | resnet56
resnet20 | resnet110
resnet20 | resnet110
resnet32 | resnet32x4
resnet8x4 | vgg13
vgg8 |
|---|---|---|---|---|---|---|---|
| KD w/ CE | 74.92 | **73.54** | 70.66 | 70.67 | 73.08 | **73.33** | 72.98 |
| **WTTM w/o CE** | **75.11** | 73.16 | **70.95** | **70.71** | **73.21** | 72.94 | **74.04** |

**Distill from better teachers.** Results in Table 7 show that the student can benefit more from a better teacher when distilling with WTTM. We observe that as the teacher model grows better, other distillation methods like KD and DIST cannot guarantee consistent improvement on the student side. In contrast, when we apply WTTM, the performance of the student is strictly increasing and consistently better than other distillation methods as the teacher becomes better and better.

Table 7: Performance of ResNet-18 on ImageNet distilled from different teachers.

| Teacher | Student | Teacher | Student | KD | DIST | **WTTM** |
|---|---|---|---|---|---|---|
| ResNet-34 | | 73.31 | | 71.21 | 72.07 | **72.19** |
| ResNet-50 | ResNet-18 | 76.13 | 69.76 | 71.35 | 72.12 | **72.26** |
| ResNet-101 | | 77.37 | | 71.09 | 72.08 | **72.34** |
| ResNet-152 | | 78.31 | | 71.12 | 72.24 | **72.39** |

**Regularization effect of TTM and WTTM.** Following our methodology, TTM and WTTM are able to embed strong regularization into the distillation process, so it's expected that student's output probability distributions $q$ resulting from TTM and WTTM should be much smoother than those resulting from KD. To validate this, we track the behavior of the average Shannon entropy of $q$ for KD, TTM and WTTM respectively during training over 3 teacher-student pairs used in CIFAR-100 experiments, shown in Fig. 1. Comparatively, students trained with TTM always have significantly larger entropy than those trained with KD. This is attributable to the Rényi entropy regularizer introduced in TTM when we remove the temperature scaling on the student side from KD. Moreover, students trained with WTTM always have slightly larger entropy than those trained with TTM, owing to the sample-adaptive weighting coefficient $U_{\frac{1}{T}}(p^t)$.

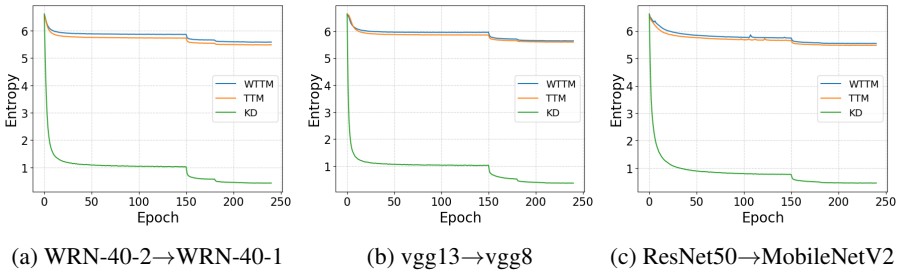

| (a) WRN-40-2→WRN-40-1 | (b) vgg13→vgg8 | (c) ResNet50→MobileNetV2 |

Figure 1: Average $H(q)$ of 3 teacher-student pairs during training. For fair comparison, we use the same temperature $T = 4$ for KD, TTM and WTTM. The $\lambda$ for KD is 0.9, so the $\beta$ for TTM is 36, computed by Eq. (19), in order to maintain the same ratio between $H(y, q)$ and $H(p_T^t, q_T)$ as KD. As for WTTM, $\beta = 36/\bar{U}$, where $\bar{U}$ is the average of $U_{\frac{1}{T}}(p^t)$ over all samples.

**WTTM facilitates more accurate teacher matching.** A closer look at TTM and WTTM is favorable to shed light on why WTTM generally performs better than TTM. To this end, we track the behavior of the average $D(p_T^t||q)$ for TTM and WTTM during training over the same 3 teacher-student pairs as above, shown in Fig. 2. In order to reflect the behavior of pure distillation, we remove $\mathcal{L}_{CE}$ from both WTTM and TTM. It's clear from the plots that WTTM always leads to smaller gap between $p_T^t$ and $q$ than TTM, demonstrating more accurate transformed teacher matching, which is the reason behind performance improvement.

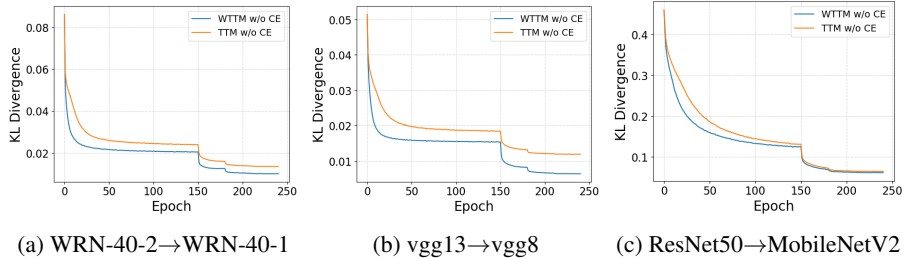

| (a) WRN-40-2→WRN-40-1 | (b) vgg13→vgg8 | (c) ResNet50→MobileNetV2 |

Figure 2: Average $D(p_T^t||q)$ of 3 teacher-student pairs during training. For each pair, the same $T$ is adopted in TTM and WTTM.

## 6 CONCLUSION

The paper systematically studies a variant of KD without temperature scaling on the student side, dubbed TTM. This slight modification gives rise to a Rényi entropy regularizer which improves the performance of the standard KD. Furthermore, we propose a sample-adaptive version of TTM, dubbed WTTM, to achieve more significant improvement. Extensive experimental results are presented to show the superiority of TTM and WTTM over other distillation methods on two image classification datasets. With almost the same training cost as KD, WTTM demonstrates state-of-the-art performance, better than most feature-based distillation methods with high computational complexity.

ACKNOWLEDGMENTS

This work was supported in part by the Natural Sciences and Engineering Research Council of Canada under Grant RGPIN203035-22, and by the Canada Research Chairs Program.

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

# A  APPENDIX

## A.1  EMPIRICAL ANALYSIS ON THE LS PERSPECTIVE OF KD

In support of our claims in Subsection 2.3, we carry out a simple empirical analysis in this section. Specifically, we train four resnet20 models on CIFAR-100 dataset with different objectives and demonstrate their Shannon entropy histograms of the output probability distributions $q$ in Figure 3.

From Figures 3(b) and 3(a), it is clear that the Shannon entropy of $q$ in the case of LS is significantly larger than its counterpart in the case of ERM, which shows the regularization effect of LS.

In comparison of Figure 3(c) with Figure 3(a), it is also clear that the Shannon entropy of $q$ in the case of KD with $T = 1$ is also significantly larger than its counterpart in the case of ERM, which confirms that KD can indeed be regarded as sample-adaptive LS when $T = 1$.

However, when $T > 1$, such a perspective doesn't hold anymore. To demonstrate this, we also trained resnet20 on CIFAR-100 dataset with KD setting $T = 4$, corresponding to Figure 3(d). Comparing Figure 3(d) with Figure 3(a), we see that the average Shannon entropy in the case of KD with $T = 4$ is even reduced over the ERM case significantly, showing an exactly opposite effect of LS. This confirms that when $T > 1$, KD can no longer be regarded as sample-adaptive LS.

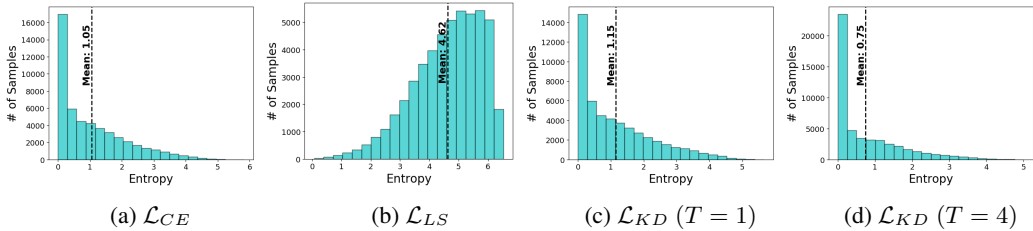

(a) $\mathcal{L}_{CE}$      (b) $\mathcal{L}_{LS}$      (c) $\mathcal{L}_{KD}$ ($T = 1$)      (d) $\mathcal{L}_{KD}$ ($T = 4$)

Figure 3: Entropy histograms for resnet20 trained with $\mathcal{L}_{CE}$, $\mathcal{L}_{LS}$ with $\epsilon = 0.5$, $\mathcal{L}_{KD}$ with $T = 1$, and $\mathcal{L}_{KD}$ with $T = 4$. For fair comparison, the same $\lambda = 0.9$ is adopted in both KD experiments with different temperatures.

## A.2  DISCUSSION ON THE GENERALIZED TRANSFORM

In this section, we provide more discussion on the generalized transform proposed in Subsection 3.1. As mentioned in Subsection 3.1, any specific transform can be described by its associated mapping $f$. For visualization, we demonstrate some examples of mapping $f$ in Fig. 4(a). Also, the power function with exponent $\gamma \in (0, 1)$ used in TTM and WTTM is visualized in Fig. 4(b).

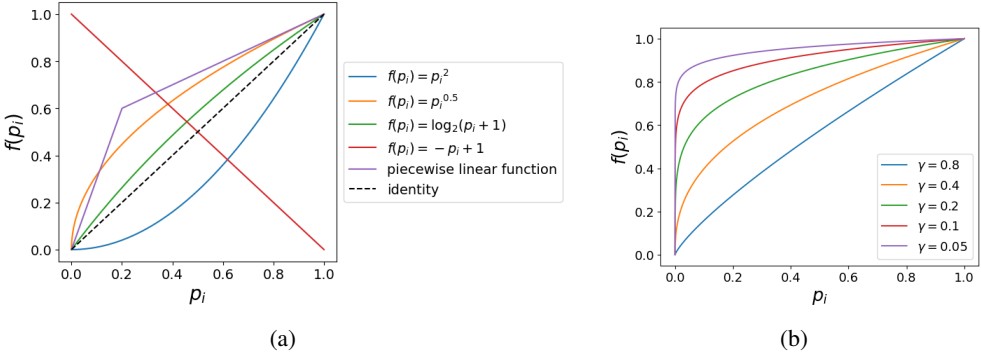

(a)                      (b)

Figure 4: (a) Various point-wise mappings. (b) Power functions with different exponents $\gamma$.

The reason why we only consider the power function in the main text is that the resulting power transform is equivalent to temperature scaling, which helps us to reveal the Rényi entropy regularizer in the subsequent derivation. However, it's worth mentioning that the generalized transform is much more than a tool used in our derivations.

Currently, we use the power transform (temperature scaling) to smooth teacher's output distributions $p$ in TTM and WTTM, following the convention in standard KD. However, it's possible that some other transforms could lead to better distillation compared to the power transform. Intuitively, mappings $f$ associated to such transforms should satisfy 3 properties:

- $f(0) = 0$ and $f(1) = 1$. A deterministic prediction shouldn't be modified by the transform.
- Non-decreasing. A non-decreasing mapping avoids ruining the order information in $p$.
- $f(p_i) > p_i$. To improve the smoothness of $p$, we need a mapping above the identity, since it expands the dynamic range of low probability values and compress the dynamic range of high probability values. As a result, after the normalization in Eq. (6), small probability values will be increased while large probability values will be decreased, achieving the goal of smoothing a distribution.

Following these suggested properties, some potential transforms can be developed in place of the power transform, while we leave this topic for future work.

### A.3 IMPLEMENTATION OF TTM AND WTTM

In this section, we provide the pseudo-code for TTM and WTTM in a Pytorch-like style, shown in Algorithm 1. It's clear that both TTM and WTTM are quite easy to implement.

---

**Algorithm 1** PyTorch-style pseudo-code for TTM and WTTM.

---

```
# y_s: student output logits
# y_t: teacher output logits
# r: the exponent for power transform

p_s = F.log_softmax(y_s, dim=1)
p_t = torch.pow(F.softmax(y_t, dim=1), r)
U = torch.sum(p_t, dim=1)    # power sum
p_t = p_t / U.unsqueeze(1)   # power transformed teacher
KL = torch.sum(F.kl_div(p_s, p_t, reduction='none'), dim=1)

# TTM
ttm_loss = torch.mean(KL)
# WTTM
wttm_loss = torch.mean(U*KL)
```

---

### A.4 HYPERPARAMETERS

We list fine-tuned $\gamma$ and $\beta$ in Tables 8, 9 and 10 covering all experiments, where $\gamma = 1/T$. Because we implement the temperature scaling with the equivalent power transform, the tuning is carried out over the exponent $\gamma$ instead of the temperature $T$.

Table 8: Hyperparameters for same-architecture distillation on CIFAR-100.

| Teacher
Student | WRN-40-2
WRN-16-2 | WRN-40-2
WRN-40-1 | resnet56
resnet20 | resnet110
resnet20 | resnet110
resnet32 | resnet32x4
resnet8x4 | vgg13
vgg8 |
|---|---|---|---|---|---|---|---|
| **TTM** | $\gamma = 0.1, \beta = 101$ | $\gamma = 0.1, \beta = 76$ | $\gamma = 0.3, \beta = 7$ | $\gamma = 0.2, \beta = 8$ | $\gamma = 0.1, \beta = 33$ | $\gamma = 0.1, \beta = 100$ | $\gamma = 0.1, \beta = 45$ |
| **WTTM** | $\gamma = 0.1, \beta = 4$ | $\gamma = 0.1, \beta = 3$ | $\gamma = 0.3, \beta = 1.5$ | $\gamma = 0.2, \beta = 2$ | $\gamma = 0.1, \beta = 1.5$ | $\gamma = 0.1, \beta = 3$ | $\gamma = 0.1, \beta = 2.25$ |
| **WTTM+CRD** | $\gamma = 0.1, \beta = 4$ | $\gamma = 0.1, \beta = 2$ | $\gamma = 0.3, \beta = 0.6$ | $\gamma = 0.2, \beta = 1.4$ | $\gamma = 0.2, \beta = 1$ | $\gamma = 0.2, \beta = 4$ | $\gamma = 0.2, \beta = 4$ |
| **WTTM+ITRD** | $\gamma = 0.3, \beta = 6$ | $\gamma = 0.4, \beta = 0.08$ | $\gamma = 0.5, \beta = 5$ | $\gamma = 0.3, \beta = 1.5$ | $\gamma = 0.3, \beta = 0.015$ | $\gamma = 0.1, \beta = 1.5$ | $\gamma = 0.1, \beta = 0.5$ |
| **WTTM** w/o CE | $\gamma = 0.2$ | $\gamma = 0.5$ | $\gamma = 0.6$ | $\gamma = 0.4$ | $\gamma = 0.4$ | $\gamma = 0.5$ | $\gamma = 0.2$ |

### A.5 COMBINATION OF DISTILLATION LOSSES

In this section, we clarify how we combine $\mathcal{L}_{WTTM}$ with other distillation losses in our experiments. Actually, we simply add another distillation component to $\mathcal{L}_{WTTM}$ with a multiplier. The total

Table 9: Hyperparameters for different-architecture distillation on CIFAR-100.

| Teacher
Student | vgg13
MobileNetV2 | ResNet50
MobileNetV2 | ResNet50
vgg8 | resnet32x4
ShuffleNetV1 | resnet32x4
ShuffleNetV2 | WRN-40-2
ShuffleNetV1 |
|---|---|---|---|---|---|---|
| **TTM** | $\gamma = 0.2, \beta = 16$ | $\gamma = 0.2, \beta = 20$ | $\gamma = 0.1, \beta = 70$ | $\gamma = 0.2, \beta = 12$ | $\gamma = 0.4, \beta = 40$ | $\gamma = 0.3, \beta = 8$ |
| **WTTM** | $\gamma = 0.2, \beta = 3$ | $\gamma = 0.2, \beta = 5$ | $\gamma = 0.1, \beta = 2$ | $\gamma = 0.2, \beta = 1.4$ | $\gamma = 0.4, \beta = 16$ | $\gamma = 0.3, \beta = 3$ |
| **WTTM+CRD** | $\gamma = 0.3, \beta = 4.2$ | $\gamma = 0.3, \beta = 3$ | $\gamma = 0.1, \beta = 3$ | $\gamma = 0.2, \beta = 0.4$ | $\gamma = 0.4, \beta = 12$ | $\gamma = 0.2, \beta = 0.16$ |
| **WTTM+ITRD** | $\gamma = 0.3, \beta = 0.03$ | $\gamma = 0.2, \beta = 0.02$ | $\gamma = 0.1, \beta = 1$ | $\gamma = 0.3, \beta = 0.6$ | $\gamma = 0.4, \beta = 0.8$ | $\gamma = 0.1, \beta = 0.2$ |

Table 10: Hyparameters for ImageNet experiments.

| Teacher | Student | **WTTM** |
|---|---|---|
| ResNet-34
ResNet-50
ResNet-101
ResNet-152 | ResNet-18 | $\gamma = 0.8, \beta = 1.6$ |
| ResNet-50 | MobileNet | $\gamma = 0.7, \beta = 3.5$ |

objective is

$$\mathcal{L}_{tot} = H(y, q) + \beta U_{\frac{1}{T}}(p^t) \cdot D(p_T^t || q) + \mu \mathcal{L}_{dist} \qquad (23)$$

where $\mu$ is a balancing weight, and $\mathcal{L}_{dist}$ is the additional distillation component, which can be CRD or ITRD in our experiments.

In the case where we combine WTTM with CRD, $\mu$ is always set to be 0.8, which is the optimal value used in the original paper.

In the case where we combine WTTM with ITRD, $\mu$ is always set to be 1. However, ITRD distillation loss itself is a combination of two components shown as follow

$$\mathcal{L}_{dist} = \beta_{corr} \mathcal{L}_{corr} + \beta_{mi} \mathcal{L}_{mi} \qquad (24)$$

where $\beta_{corr}$ and $\beta_{mi}$ are two balancing weights within ITRD distillation loss. In our experiments, we always select the optimal $\beta_{corr}$ and $\beta_{mi}$ values specified in the original paper. Specifically, $\beta_{corr} = 2$ and $\beta_{mi} = 0$ for 3 teacher-student pairs, namely ResNet50 $\rightarrow$ MobileNetV2, ResNet50 $\rightarrow$ vgg8 and WRN-40-2 $\rightarrow$ ShuffleNetV1, while $\beta_{corr} = 2$ and $\beta_{mi} = 1$ for all the other 10 teacher-student pairs. Note that there is another inherent hyperparameter $\alpha_{it}$ within ITRD, which is selected as 1.01 for same-architecture distillation and 1.5 for different-architecture distillation, following the suggestion in the original paper.

## A.6 FUTURE WORK

This work provides multiple directions for our future research:

- From Eq. (15), we know that the ratio between the distillation term $D(p_T^t || q_T)$ and the regularizer $H_{\frac{1}{T}}(q)$ in TTM is determined by $T$. Also, the order of Rényi entropy is bound to be $1/T$. However, these constraints are not necessary. In future work, we can directly combine the standard KD with a Rényi entropy regularizer while setting the balancing weight and the order of Rényi entropy as tunable hyperparameters.

- Given the generalized transform framework and related discussion in A.2, other transforms can be proposed in place of the power transform (temperature scaling) used in TTM and WTTM.

- Systematically analyze the selection of the sample-adaptive weight in WTTM, in order to find the optimal one.

## A.7 RELATED WORK

In recent years, a variety of works have been proposed to advance the methodology of KD and its application to related fields. Huang et al. (2022) proposed a correlation-based loss capturing the inter-class and intra-class relations from the teacher explicitly. Yang et al. (2023b) unified KD and self distillation by decomposing and reorganizing the vanilla KD loss into a normalized KD (NKD) loss and proposed a novel self distillation method based on it. Li et al. (2023c) proposed a novel distillation method based on a dynamic and learnable distillation temperature. Hao et al. (2023) claimed that the power of vanilla KD was underestimated due to small data pitfall, and observed that the performance gap between vanilla KD and other meticulously designed KD variants could be greatly reduced by employing stronger training strategy. Li (2022) proposed a novel feature-based self distillation approach, reusing channel-wise and layer-wise features within the student to provide regularization. Liu et al. (2023) presented a two-stage KD method dubbed NORM based on a feature transform module. Li & Jin (2022) proposed a Shadow Knowledge Distillation framework to bridge offline and online distillation in an efficient way. Dong et al. (2023) presented a training-free framework to search for the optimal student architectures given a teacher architecture. Also, following the trend of Automated Machine Learning (AutoML), several recent works (Li et al., 2023a;b) focused on automating distiller design using techniques like evolutionary algorithm and Monte Carlo tree search.

## A.8 STANDARD DEVIATION FOR RESULTS ON CIFAR-100

Below, we report the standard deviation for results on CIFAR-100 dataset in Table 11 and 12.

Table 11: Top-1 accuracy (%) on CIFAR-100. Each teacher-student pair has the **same** architecture. Standard deviation is provided (the standard deviation is missing for DKD since it's not available in the literature).

| Teacher
Student | WRN-40-2
WRN-16-2 | WRN-40-2
WRN-40-1 | resnet56
resnet20 | resnet110
resnet20 | resnet110
resnet32 | resnet32x4
resnet8x4 | vgg13
vgg8 |
|---|---|---|---|---|---|---|---|
| Teacher | 75.61 | 75.61 | 72.34 | 74.31 | 74.31 | 79.42 | 74.64 |
| Student | 73.26 | 71.98 | 69.06 | 69.06 | 71.14 | 72.50 | 70.36 |
| *Feature-based* | | | | | | | |
| FitNet | $73.58 \pm 0.32$ | $72.24 \pm 0.24$ | $69.21 \pm 0.36$ | $68.99 \pm 0.27$ | $71.06 \pm 0.13$ | $73.50 \pm 0.28$ | $71.02 \pm 0.31$ |
| AT | $74.08 \pm 0.25$ | $72.77 \pm 0.10$ | $70.55 \pm 0.27$ | $70.22 \pm 0.16$ | $72.31 \pm 0.08$ | $73.44 \pm 0.19$ | $71.43 \pm 0.09$ |
| VID | $74.11 \pm 0.24$ | $73.30 \pm 0.13$ | $70.38 \pm 0.14$ | $70.16 \pm 0.39$ | $72.61 \pm 0.28$ | $73.09 \pm 0.21$ | $71.23 \pm 0.06$ |
| RKD | $73.35 \pm 0.09$ | $72.22 \pm 0.20$ | $69.61 \pm 0.06$ | $69.25 \pm 0.05$ | $71.82 \pm 0.34$ | $71.90 \pm 0.11$ | $71.48 \pm 0.05$ |
| PKT | $74.54 \pm 0.04$ | $73.45 \pm 0.19$ | $70.34 \pm 0.04$ | $70.25 \pm 0.04$ | $72.61 \pm 0.17$ | $73.64 \pm 0.18$ | $72.88 \pm 0.09$ |
| CRD | $75.48 \pm 0.09$ | $74.14 \pm 0.22$ | $71.16 \pm 0.17$ | $71.46 \pm 0.09$ | $73.48 \pm 0.13$ | $75.51 \pm 0.18$ | $73.94 \pm 0.22$ |
| *Logits-based* | | | | | | | |
| KD | $74.92 \pm 0.28$ | $73.54 \pm 0.20$ | $70.66 \pm 0.24$ | $70.67 \pm 0.27$ | $73.08 \pm 0.18$ | $73.33 \pm 0.25$ | $72.98 \pm 0.19$ |
| DIST | $75.51 \pm 0.04$ | $74.73 \pm 0.24$ | $71.75 \pm 0.30$ | $71.65 \pm 0.21$ | $73.69 \pm 0.23$ | $76.31 \pm 0.19$ | $73.89 \pm 0.19$ |
| DKD | 76.24 | 74.81 | 71.97 | n/a | 74.11 | 76.32 | 74.68 |
| **TTM** | $76.23 \pm 0.15$ | $74.32 \pm 0.31$ | $71.83 \pm 0.16$ | $71.46 \pm 0.16$ | $73.97 \pm 0.23$ | $76.17 \pm 0.28$ | $74.33 \pm 0.07$ |
| **WTTM** | $76.37 \pm 0.10$ | $74.58 \pm 0.26$ | $71.92 \pm 0.40$ | $71.67 \pm 0.28$ | $74.13 \pm 0.37$ | $76.06 \pm 0.27$ | $74.44 \pm 0.19$ |
| **WTTM+CRD** | $76.61 \pm 0.24$ | $74.94 \pm 0.35$ | $72.20 \pm 0.15$ | $72.13 \pm 0.26$ | $74.52 \pm 0.29$ | $76.65 \pm 0.14$ | $74.71 \pm 0.07$ |
| **WTTM+ITRD** | $76.65 \pm 0.33$ | $75.34 \pm 0.22$ | $72.16 \pm 0.28$ | $72.20 \pm 0.27$ | $74.36 \pm 0.31$ | $77.36 \pm 0.13$ | $75.13 \pm 0.16$ |

## A.9 RESULTS ON TRANSFORMER-BASED MODELS

To verify the effectiveness of our proposed distillation method WTTM on transformer-based models, we apply it to a vision transformer model DeiT-Tiny (Touvron et al., 2021), results shown in Table 13. We conduct experiments following the settings in Yang et al. (2023b) and Yang et al. (2022), and compare our results with the vanilla KD and two distillation methods proposed in the above two papers, namely NKD and ViTKD. It's shown that the performance of WTTM is better than all the three benchmark methods. Moreover, combined with ViTKD, WTTM can improve the Top-1 accuracy of DeiT-Tiny to 78.04%, which is also higher than the performance of NKD combined with ViTKD.

Table 12: Top-1 accuracy (%) on CIFAR-100. Each teacher-student pair has **different** architectures. Standard deviation is provided (the standard deviation is missing for DKD since it's not available in the literature).

| Teacher
Student | vgg13
MobileNetV2 | ResNet50
MobileNetV2 | ResNet50
vgg8 | resnet32x4
ShuffleNetV1 | resnet32x4
ShuffleNetV2 | WRN-40-2
ShuffleNetV1 |
|---|---|---|---|---|---|---|
| Teacher | 74.64 | 79.34 | 79.34 | 79.42 | 79.42 | 75.61 |
| Student | 64.6 | 64.6 | 70.36 | 70.5 | 71.82 | 70.5 |
| *Feature-based* | | | | | | |
| FitNet | $64.14 \pm 0.50$ | $63.16 \pm 0.47$ | $70.69 \pm 0.22$ | $73.59 \pm 0.15$ | $73.54 \pm 0.22$ | $73.73 \pm 0.32$ |
| AT | $59.40 \pm 0.20$ | $58.58 \pm 0.54$ | $71.84 \pm 0.28$ | $71.73 \pm 0.31$ | $72.73 \pm 0.09$ | $73.32 \pm 0.35$ |
| VID | $65.56 \pm 0.42$ | $67.57 \pm 0.28$ | $70.30 \pm 0.31$ | $73.38 \pm 0.09$ | $73.40 \pm 0.17$ | $73.61 \pm 0.12$ |
| RKD | $64.52 \pm 0.45$ | $64.43 \pm 0.42$ | $71.50 \pm 0.07$ | $72.28 \pm 0.39$ | $73.21 \pm 0.28$ | $72.21 \pm 0.16$ |
| PKT | $67.13 \pm 0.30$ | $66.52 \pm 0.33$ | $73.01 \pm 0.14$ | $74.10 \pm 0.25$ | $74.69 \pm 0.34$ | $73.89 \pm 0.16$ |
| CRD | $69.73 \pm 0.42$ | $69.11 \pm 0.28$ | $74.30 \pm 0.14$ | $75.11 \pm 0.32$ | $75.65 \pm 0.10$ | $76.05 \pm 0.14$ |
| *Logits-based* | | | | | | |
| KD | $67.37 \pm 0.32$ | $67.35 \pm 0.32$ | $73.81 \pm 0.13$ | $74.07 \pm 0.19$ | $74.45 \pm 0.27$ | $74.83 \pm 0.17$ |
| DIST | $68.50 \pm 0.26$ | $68.66 \pm 0.23$ | $74.11 \pm 0.07$ | $76.34 \pm 0.18$ | $77.35 \pm 0.25$ | $76.40 \pm 0.03$ |
| DKD | 69.71 | 70.35 | n/a | 76.45 | 77.07 | 76.70 |
| **TTM** | $68.98 \pm 0.85$ | $69.24 \pm 0.28$ | $74.87 \pm 0.31$ | $74.18 \pm 0.26$ | $76.57 \pm 0.26$ | $75.39 \pm 0.33$ |
| **WTTM** | $69.16 \pm 0.20$ | $69.59 \pm 0.58$ | $74.82 \pm 0.28$ | $74.37 \pm 0.39$ | $76.55 \pm 0.08$ | $75.42 \pm 0.34$ |
| **WTTM**+CRD | $70.30 \pm 0.68$ | $70.84 \pm 0.56$ | $75.30 \pm 0.42$ | $75.82 \pm 0.16$ | $77.04 \pm 0.19$ | $76.86 \pm 0.37$ |
| **WTTM**+ITRD | $70.70 \pm 0.45$ | $71.56 \pm 0.15$ | $76.00 \pm 0.17$ | $77.03 \pm 0.26$ | $77.68 \pm 0.26$ | $77.44 \pm 0.27$ |

Table 13: Top-1 accuracy (%) on ImageNet.

| Teacher | Student | KD | ViTKD | NKD | **WTTM** | NKD+ViTKD | **WTTM+ViTKD** |
|---|---|---|---|---|---|---|---|
| DeiT III-Small (82.76) | DeiT-Tiny (74.42) | 76.01 | 76.06 | 76.68 | 77.03 | 77.78 | 78.04 |

