# OpenReview forum: "Knowledge Distillation Based on Transformed Teacher Matching"
_ICLR.cc/2024/Conference — ICLR 2024 poster_

### Official Review · Reviewer_fD3s · 2023-10-13

**Soundness:** 2 fair
**Presentation:** 4 excellent
**Contribution:** 2 fair
**Rating:** 8
**Confidence:** 5

**Summary:**

This paper proposes a new variant of knowledge distillation called Transformed Teacher Matching (TTM) that drops temperature scaling on the student side and introduces an inherent regularization term. The paper shows that TTM leads to better generalization and achieves state-of-the-art accuracy performance. The paper also introduces a weighted version of TTM called Weighted Transformed Teacher Matching (WTTM) that enhances the student's capability to match the teacher's power transformed probability distribution. The experiments conducted in the paper demonstrate the effectiveness of TTM and WTTM on various datasets and architectures.

**Strengths:**

1. The paper introduces a new variant of knowledge distillation that drops temperature scaling on the student side and introduces an inherent regularization term. This approach is motivated by recent works and is a departure from conventional knowledge distillation. The paper also introduces a weighted version of TTM that enhances the student's capability to match the teacher's power transformed probability distribution. These contributions are novel and have not been explored in previous works.

2. The paper is well-written and presents a clear and concise description of the proposed methods. The authors provide a thorough analysis of the experimental results and compare their approach with state-of-the-art methods. The experiments are conducted on various datasets and architectures, which demonstrates the effectiveness and robustness of the proposed methods.

3. The proposed methods have the potential to improve the performance of knowledge distillation and have practical applications in various domains. The paper demonstrates that TTM and WTTM achieve state-of-the-art accuracy performance on various datasets and architectures. The inherent regularization term in TTM also provides a new perspective on knowledge distillation and has the potential to inspire further research in this area. Overall, the paper makes a significant contribution to the field of knowledge distillation

**Weaknesses:**

1. The paper could benefit from addressing the lack of novelty by acknowledging that techniques such as R´enyi or f divergence, temperature scaling, and logits normalization have already been widely used in knowledge distillation. For example, Information Theoretic Representation Distillation (BMVC) employed R´enyi divergence for standard distillation, and AlphaNet (ICML2021) utilized the f divergence to distill different sub-networks. Moreover, this method is likely already considered in the distiller's search work (KD-Zero: Evolving Knowledge Distiller for Any Teacher-Student Pairs, NeurIPS-2023).

2. To strengthen the paper's findings, it is important to validate the proposed method on downstream tasks such as object detection and segmentation. Including evaluation results on these tasks will demonstrate the practical effectiveness and applicability of the proposed method. Additionally, providing more examples and visualizations will enhance the readers' understanding of how the method works and its impact on the learning process.

3. Furthermore, it is essential to incorporate a thorough discussion of relevant KD-related studies, including Self-Regulated Feature Learning via Teacher-free Feature Distillation (ECCV2022), NORM: Knowledge Distillation via N-to-One Representation Matching (ICLR2023), Shadow Knowledge Distillation: Bridging Offline and Online Knowledge Transfer (NIPS2022), DisWOT: Student Architecture Search for Distillation Without Training (CVPR2023), and Automated Knowledge Distillation via Monte Carlo Tree Search (ICCV2023). This discussion will help position the proposed approach within the existing literature, establish connections, and provide valuable insights for potential comparisons.

**Questions:**

The only concern to me is the novelty of the work and I hope the authors could discuss some of the related work I mentioned in the revised version.


---------------------------------

The author's response addressed my concerns well, so I'm improving my score to acceptance, thanks!

---

> ### Author Response · Authors · 2023-11-18
> **Response to Reviewer fD3s (Part 1)**
>
> We thank you very much for taking time to review our paper and provide valuable feedbacks. To address your major concern about the novelty of our work, we have discussed the related works you mentioned in the revised version of our paper (see appendix A.7). Please find our point-by-point responses to your comments as follows.
>
> > Comment 1: The paper could benefit from addressing the lack of novelty by acknowledging that techniques such as R´enyi or f divergence, temperature scaling, and logits normalization have already been widely used in knowledge distillation. For example, Information Theoretic Representation Distillation (BMVC) employed R´enyi divergence for standard distillation, and AlphaNet (ICML2021) utilized the f divergence to distill different sub-networks. Moreover, this method is likely already considered in the distiller's search work (KD-Zero: Evolving Knowledge Distiller for Any Teacher-Student Pairs, NeurIPS-2023).
>
> **Response**: One of the major contributions of our work is showing that dropping the temperature on the student side leads to a better version of KD, dubbed as TTM in the paper. In our work, R´enyi entropy is not something deliberately involved by us to build our own method. Instead, it naturally emerges when we remove the temperature on the student side, which explains why dropping the temperature on the student side is beneficial. Therefore, although similar concepts appear in both our work and the works you listed, the context is totally different.
>
> To further alleviate your concern, we next show that how our work is different from the works you mentioned one by one as follows: (1) Information Theoretic Representation Distillation is a feature-based distillation method, while our proposed method is logits-based, with only minimal modifications to the original KD. (2) AlphaNet utilizes $\alpha$-divergence in distillation instead of the conventional KL divergence. However, we stick to the KL divergence used in the original KD. (3) We acknowledge that KD-Zero does search for distillers in a pretty wide searching space. However, according to Table 1 in the paper of KD-Zero, we firmly believe that the adaptive weighting term $U$ proposed in our work is not considered in the searching space of KD-Zero.
>
> Actually, based on our literature review, the most similar work to us is Self-Distillation as Instance-Specific Label Smoothing (NIPS 2020), and we have already carefully addressed this work in Subsection 2.3.
>
> Based on the above discussion, we have full confidence in the novelty of our work.
>
> > Comment 2: To strengthen the paper's findings, it is important to validate the proposed method on downstream tasks such as object detection and segmentation. Including evaluation results on these tasks will demonstrate the practical effectiveness and applicability of the proposed method. Additionally, providing more examples and visualizations will enhance the readers' understanding of how the method works and its impact on the learning process.
>
> **Response**: The major contribution of this paper is to show that it is better off to drop the temperature T from the student side. This is well supported by both theoretical analysis and empirical results. This is a new understanding about KD and, together with the statistical perspective of KD, offers a new explanation of why KD helps. On top of this contribution, we further introduce a sample-adaptive coefficient to KD, yielding the method called WTTM. While TTM can be regarded as a correction of KD, WTTM is new and achieves the state-of-the-art performance. In addition, WTTM has essentially the same computation complexity as KD does. It can be applied wherever KD is applicable.  In the literature, KD has already been widely used on downstream tasks such as object detection and segmentation. So, there’s no reason to believe our proposed methods are not applicable to them. Of course, how much gain TTM and WTTM would offer in comparison with KD and other existing distillers on those tasks needs to be found out in future work.
>
> As for examples and visualizations, please refer to Figure 1 in our paper, where we visualize the regularization effect of our proposed objectives compared to vanilla KD, and Figure 2, where we visualize the behavior of the average $D(p^t_T||q)$ for TTM and WTTM throughout the learning process, which helps explain why WTTM performs better than TTM.

---

> > ### Comment · Reviewer_fD3s · 2023-11-18
> > **Thanks for the response.**
> >
> > The author's response addressed my concerns well, so I'm improving my score to acceptance, thanks!

---

> > > ### Author Response · Authors · 2023-11-18
> > > **Thanks for Your Appreciation of Our Responses**
> > >
> > > Thank you so much for your appreciation of our responses. We are glad that you raised your score. Many thanks to your thorough comments which helped us to improve our work.

---

> ### Author Response · Authors · 2023-11-18
> **Response to Reviewer fD3s (Part 2)**
>
> > Comment 3: Furthermore, it is essential to incorporate a thorough discussion of relevant KD-related studies, including Self-Regulated Feature Learning via Teacher-free Feature Distillation (ECCV2022), NORM: Knowledge Distillation via N-to-One Representation Matching (ICLR2023), Shadow Knowledge Distillation: Bridging Offline and Online Knowledge Transfer (NIPS2022), DisWOT: Student Architecture Search for Distillation Without Training (CVPR2023), and Automated Knowledge Distillation via Monte Carlo Tree Search (ICCV2023). This discussion will help position the proposed approach within the existing literature, establish connections, and provide valuable insights for potential comparisons.
>
> **Response**: Please refer to appendix A.7 of our revised paper, where we cite all these papers you mentioned and provide a thorough discussion of relevant KD-related studies.

---

### Official Review · Reviewer_2Z26 · 2023-10-30

**Soundness:** 2 fair
**Presentation:** 3 good
**Contribution:** 2 fair
**Rating:** 6
**Confidence:** 4

**Summary:**

This paper investigates the temperature for Knowledge Distillation, proposing Transformed Teacher Matching (TTM), which drops the temperature scaling on the student side. TTM has an inherent Renyi entropy term in its objective function, and this regularization leads to better performance with KD.

**Strengths:**

1. The method that rethinking KD via temperature scaling is interesting.
2. The final TTM does not introduce extra hyper-parameters. Also, the training speed keeps the same.
3. The results on various datasets and models prove its effectiveness.

**Weaknesses:**

1. Some references and comparisons are missing:

    [1] Knowledge distillation from a stronger teacher.

    [2] From Knowledge Distillation to Self-Knowledge Distillation: A Unified Approach with Normalized Loss and Customized Soft Labels.

    [3] Curriculum Temperature for Knowledge Distillation.

    [4] VanillaKD: Revisit the Power of Vanilla Knowledge Distillation from Small Scale to Large Scale.
2. When temperature=1, is TTM the same as the original KD? In some papers, the temperature on ImageNet is actually 1.0.
3. Could TTM still achieve better performance for larger models (e.g. DeiT-T or DeiT-S)?  VanillaKD shows under strong training settings, the original KD also performs well.

**Questions:**

above

---

> ### Author Response · Authors · 2023-11-18
> **Response to Reviewer 2Z26**
>
> We thank you very much for taking time to review our paper and provide valuable feedbacks. We have addressed the papers you mentioned in the revised version of our paper (see appendix A.7). Also, we managed to generate some preliminary result for DeiT-T to address your concern. Below, please find our point-by-point responses to your comments.
>
> > Comment 1: Some references and comparisons are missing …
>
> **Response**: Please refer to appendix A.7 in our revised paper, where we include the 4 papers you mentioned as references. As for comparison, in the revised version, we include the first 3 papers you mentioned as benchmarks in Table 5, and our method is still achieving outstanding performance. However, there are no overlapping experimental settings between our work and VanillaKD to form a fair comparison, and it's infeasible for us to follow the settings in VanillaKD to generate some new results in such limited time based on our computational resources, so we didn't include VanillaKD into comparison.
>
> > Comment 2: When temperature=1, is TTM the same as the original KD? In some papers, the temperature on ImageNet is actually 1.0.
>
> **Response**: Yes, TTM (and WTTM) is the same as the original KD when temperature=1, and we know that the commonly used temperature on ImageNet is 1.0 for the original KD. However, with temperature dropped on the student side, the effect of the temperature in distillation is changed, so it's reasonable for TTM (and WTTM) to have a different optimal temperature from the original KD. According to our experiments, the optimal $\gamma$ for our method is around 0.8 on ImageNet (see Table 10 in our revised paper), which is equivalent to temperature=1.25. Consequently, our method is different from the original KD on ImageNet.
>
> > Comment 3: Could TTM still achieve better performance for larger models (e.g. DeiT-T or DeiT-S)? VanillaKD shows under strong training settings, the original KD also performs well.
>
> **Response**: Both TTM and WTTM have essentially the same computation complexity as KD does. They can be applied wherever KD is applicable. In the literature, KD has already been widely used in improving the performance of larger models such as the transformer models you mentioned. So, there’s no reason why TTM and WTTM cannot.
>
> In the interest of time and also due to our limited computational resources, we tried our best in the last a couple of days and managed to generate one result for DeiT-T on ImageNet without tuning any hyperparameters. The result is shown in the table below.
>
> |Teacher|Student|KD|WTTM|
> |:---:|:---:|:---:|:---:|
> |RegNetY-16GF (82.9) |DeiT-T (72.2)|72.2|72.7|
>
> Following the training strategy of the original DeiT paper [1], KD is not able to improve the Top-1 accuracy (%) performance of DeiT-T on ImageNet (72.2%), while WTTM can improve the performance to 72.7%. This small example at least shows the potential of WTTM for larger models.
>
> Note that the training strategy in [1] is also quite strong, including AdamW optimizer, cosine LR decay, label smoothing, stochastic depth, repeated augmentation, Rand Augment, mixup, cutmix, random erasing and so on, with carefully tuned hyperparameters for each ingredient. Therefore, we believe that our method is still better than the original KD even in strong training settings.
>
> [1] Touvron et al., "Training data-efficient image transformers & distillation through attention", ICML 2021.

---

> > ### Comment · Reviewer_2Z26 · 2023-11-20
> >
> > I am still concerned about the perfromance of larger models. In Tab.11 of paper NKD, KD can improve the performance of DeiT actually. Could you provide the performance with the proposed WTTM? It may need about two days to train a DeiT-Tiny.

---

> > > ### Author Response · Authors · 2023-11-22
> > > **Additional Result**
> > >
> > > Thank you for your reply. In the past two days, we have conducted the experiment for WTTM using the official implementation of the NKD [1] paper. We directly used the configuration file specified in the official repository and simply replaced the code of the NKD loss with our implementation of the WTTM loss. For reproducibility, we included the files we modified in our supplementary materials (see folder TTM_deit), together with the log file of our experiment. As you may find out in the log file, our experiment stopped at the end of the 273rd epoch unexpectedly, due to some file system error. However, we didn't have enough time to rerun the experiment, so we decided to report the performance at the 272nd epoch.
> > >
> > > Our result is shown in the following table:
> > >
> > > |Teacher|Student|KD|NKD|WTTM|
> > > |:---:|:---:|:---:|:---:|:---:|
> > > |DeiT III-Small (82.76)|DeiT-Tiny (74.42)|76.01|76.68|77.69|
> > >
> > > It's clear that WTTM has an outstanding performance even though we didn't finish all the 300 epochs, and we believe that the performance of WTTM can be further improved in the last 28 epochs.
> > >
> > > Hopefully, this result can address your concern.
> > >
> > > [1] Yang et al., "From Knowledge Distillation to Self-Knowledge Distillation: A Unified Approach with Normalized Loss and Customized Soft Labels", ICCV 2023.

---

> ### Author Response · Authors · 2023-11-22
>
> Dear Reviewer 2Z26,
>
> Thanks again for reviewing our paper. Since the discussion period is going to end soon, we are eager to know if you are satisfied with our previous responses. If no, please kindly tell us your remaining concerns and hopefully we can address them before the deadline. If yes, we wonder if it’s possible for you to raise the score. In any case, we would be extremely grateful to hear from you.
>
> Thanks.

---

> > ### Comment · Reviewer_2Z26 · 2023-11-23
> >
> > I suggest to add the results for DeiT. I decide to increase my score.

---

> > > ### Author Response · Authors · 2023-11-23
> > > **Thanks for Your Appreciation of Our Responses**
> > >
> > > Thank you so much for your appreciation of our responses. We are glad that you raised your score.
> > > We will rerun the experiment to complete 300 epochs and add our results for Deit in the final version of our paper.

---

### Official Review · Reviewer_DiN5 · 2023-10-31

**Soundness:** 3 good
**Presentation:** 3 good
**Contribution:** 2 fair
**Rating:** 6
**Confidence:** 3

**Summary:**

The paper systematically studies a variant of KD without temperature scaling on the student side, dubbed TTM. Temperature scaling is crucial in knowledge distillation (KD). This paper introduces transformed teacher matching (TTM), a variant of KD that omits temperature scaling on the student side. TTM includes an inherent regularization term and produces better generalization compared to the original KD. Weighted TTM (WTTM) further enhances the student's ability to match the teacher's probability distribution, achieving state-of-the-art accuracy.

**Strengths:**

- Fruitful discussion about related works to engage the readers.
- Theoretical derivation from KD to the proposed TTM.

**Weaknesses:**

The results are completely dependent on the list T and β values of all experiments (see Table 8 and 9), which makes the method impractical. Furthermore, the optimal value may even vary from task to task, dataset to dataset and backbone to backbone. These are my main concerns. Based on the marginal gain compared to the baselines, these empirical results actually weaken the claimed contribution.

**Questions:**

see above

---

> ### Author Response · Authors · 2023-11-18
> **Response to Reviewer DiN5**
>
> We thank you very much for taking time to review our paper and provide valuable feedbacks. Please find our response to your comment below.
>
> > Comment: The results are completely dependent on the list T and β values of all experiments (see Table 8 and 9), which makes the method impractical. Furthermore, the optimal value may even vary from task to task, dataset to dataset and backbone to backbone. These are my main concerns. Based on the marginal gain compared to the baselines, these empirical results actually weaken the claimed contribution.
>
> **Response**: We appreciate your concerns mentioned above and will address them from four different perspectives.
>
> - First, the major contribution of this paper is to show that it is better off to drop the temperature T from the student side. This is well supported by both theoretical analysis and empirical results. Regardless of how large or small the gain is, the point is that removing the temperature from the student side leads to a better distiller with more regularization. This is a new understanding about KD and, together with the statistical perspective of KD, offers a new explanation of why KD helps. On top of this contribution, we further introduce a sample-adaptive coefficient to KD, yielding the method called WTTM. While TTM can be regarded as a correction of KD, WTTM is new and achieves the state-of-the-art performance.
> - Second, as for practicality, WTTM has essentially the same computation complexity as KD does. It can be applied wherever KD is applicable. Third, yes, for results in Table 1, hyperparameters are tuned according to Tables 8 and 9. However, this is done for the purpose of fair comparison and benchmarking since some benchmark method (such as DKD [1]) also has its hyperparameters tuned from task to task, dataset to dataset and backbone to backbone. In addition, if one examines Tables 8 and 9 carefully, it can be seen that hyperparameters of WTTM do not fluctuate much from one teacher-student pair to another in most cases. This implies that even with fixed hyperparameters, WTTM would perform well in general for all tested teacher-student pairs.
> - Fourth, to further confirm the point mentioned above, the table below shows the results (Top-1 accuracy, %) of KD and WTTM on CIFAR-100 with $T=4$ and $\beta=36/\bar{U}$ fixed for all teacher-student pairs, the same as how we generate Figure 1 in our paper. Note that these parameters are optimized for KD, but not for WTTM. From the table below, it is clear that WTTM still outperforms KD by a large margin in most cases.
>
> ||KD|WTTM|
> |:---:|:---:|:---:|
> |WRN-40-2 &#8594; WRN-16-2|74.92|75.88|
> |WRN-40-2 &#8594; WRN-40-1|73.54|74.17|
> |resnet56 &#8594; resnet20|70.66|71.52|
> |resnet110 &#8594; resnet20|70.67|71.18|
> |resnet110 &#8594; resnet32|73.08|73.49|
> |resnet32x4 &#8594; resnet8x4|73.33|74.97|
> |vgg13 &#8594; vgg8|72.98|73.66|
> |vgg13 &#8594; MobileNetV2|67.37|68.12|
> |ResNet50 &#8594; MobileNetV2|67.35|68.59|
> |ResNet50 &#8594; vgg8|73.81|74.13|
> |resnet32x4 &#8594; ShuffleNetV1|74.07|73.45|
> |resnet32x4 &#8594; ShuffleNetV2|74.45|76.13|
> |WRN-40-2 &#8594; ShuffleNetV1|74.83|74.81|
>
> [1] Zhao et al., "Decoupled knowledge distillation", CVPR 2022.

---

> ### Author Response · Authors · 2023-11-22
>
> Dear Reviewer DiN5,
>
> Thanks again for reviewing our paper. Since the discussion period is going to end soon, we are eager to know if you are satisfied with our previous response. If no, please kindly tell us your remaining concerns and hopefully we can address them before the deadline. If yes, we wonder if it’s possible for you to raise the score. In any case, we would be extremely grateful to hear from you.
>
> Thanks.

---

> ### Author Response · Authors · 2023-11-23
>
> Dear Reviewer DiN5, the discussion period is going to end within several hours, and we are still eager to know your position after our previous response. If it addressed your concern, we wonder if it’s possible for you to raise the score. We would be extremely grateful to hear from you.
>
> Thanks.

---

### Official Review · Reviewer_LeDY · 2023-11-01

**Soundness:** 4 excellent
**Presentation:** 2 fair
**Contribution:** 3 good
**Rating:** 6
**Confidence:** 2

**Summary:**

The paper systematically analyzed the effect of dropping the temperature scaling on the student side in knowledge distillation (KD). The theoretical analysis shows that such a transformation leads to a general KD loss and a Renyi entropy regularization that improves the generalization of the student. Further, To further enhance student’s capability to match teacher’s power transformed probability distribution, the paper introduces a sample-adaptive coefficient to the method. Experiments are conducted to validate the effectiveness of both modules. Experiments are evaluated with different model architectures and teacher quality.

**Strengths:**

I think overall the paper provides new findings to understand the role of temperature in knowledge distillation. And the evaluation experiments are extensive.

1. The theoretical derivation and analysis for the general KD, Renyi entropy, and transformed teacher matching is precise and solid.

2. Extensive experiments confirm the theoretical analysis and show the effectiveness of each proposed module.

**Weaknesses:**

1. It's better to provide a detailed summary and comparison of the latest related works.

2. It's also more convincing to show results on transformer models such as ViT.

**Questions:**

Please see the weakness part.

---

> ### Author Response · Authors · 2023-11-18
> **Response to Reviewer LeDY**
>
> We thank you very much for taking time to review our paper and provide valuable feedbacks. We have added a subsection in our paper (see appendix A.7) to summarize some latest related works about KD, and compared our results with more latest works in the result section (see Table 5) as well. Also, we managed to generate some preliminary result on a transformer model. Below, please find our point-by-point responses to your comments.
>
> > Comment 1: It's better to provide a detailed summary and comparison of the latest related works.
>
> **Response**: Please refer to appendix A.7 in our revised paper, where we provide a summary of the latest related works. As for comparison, in the revised version, we include more benchmarks in Table 5 including many works published last year and this year, which are DKD [1], DIST [2], KD++ [3], NKD [4], CTKD [5], and KD-Zero [6], and our method is still achieving outstanding performance.
>
> [1] Zhao et al., "Decoupled knowledge distillation", CVPR 2022.
> [2] Huang et al., "Knowledge distillation from a stronger teacher", NIPS 2022.
> [3] Wang et al., "Improving knowledge distillation via regularizing feature norm and direction", Under review of ICLR 2024.
> [4] Yang et al., "From Knowledge Distillation to Self-Knowledge Distillation: A Unified Approach with Normalized Loss and Customized Soft Labels", ICCV 2023.
> [5] Li et al., "Curriculum Temperature for Knowledge Distillation", AAAI 2023.
> [6] Li et al., "KD-Zero: Evolving Knowledge Distiller for Any Teacher-Student Pairs", NIPS 2023.
>
> > Comment 2: It's also more convincing to show results on transformer models such as ViT.
>
> **Response**: Both TTM and WTTM have essentially the same computation complexity as KD does. They can be applied wherever KD is applicable. In the literature, KD has already been widely used in improving the performance of transformer models. So, there’s no reason why TTM and WTTM cannot.
>
> In the interest of time and also due to our limited computational resources, we tried our best in the last a couple of days and managed to generate one result for one of the smallest transformer models, namely DeiT-tiny, on ImageNet without tuning any hyperparameters. The result is shown in the table below.
>
> |Teacher|Student|KD|WTTM|
> |:---:|:---:|:---:|:---:|
> |RegNetY-16GF (82.9) |DeiT-tiny (72.2)|72.2|72.7|
>
> Following the training strategy of the original DeiT paper [7], KD is not able to improve the Top-1 accuracy (%) performance of DeiT-tiny on ImageNet (72.2%), while WTTM can improve the performance to 72.7%. This small example at least shows the potential of WTTM for transformer models.
>
> [7] Touvron et al., "Training data-efficient image transformers & distillation through attention", ICML 2021.

---

> ### Author Response · Authors · 2023-11-22
>
> Dear Reviewer LeDY,
>
> Thanks again for reviewing our paper. We have more result about transformer models. Please refer to our latest response to Reviewer 2Z26.
>
> Since the discussion period is going to end soon, we are eager to know if you are satisfied with our previous response. If no, please kindly tell us your remaining concerns and hopefully we can address them before the deadline. If yes, we wonder if it’s possible for you to raise the score. In any case, we would be extremely grateful to hear from you.
>
> Thanks.

---

> ### Author Response · Authors · 2023-11-23
>
> Dear Reviewer LeDY, the discussion period is going to end within several hours, and we are still eager to know your position after our previous response. If it addressed your concern, we wonder if it’s possible for you to raise the score. We would be extremely grateful to hear from you.
>
> Thanks.

---

### Public Comment · ~Peter_Chen5 · 2023-11-20
**variance of results**

Dear author,

We are interested in your work. However, after we attempting to reproduce the results in table 2 and table 3, using your provided code, we realized that WTTM is very sensitive to hyper-parameters T and beta, i.e., a change in these two values changes student's accuracy a lot. Also, the variance of student accuracy for the VGG13-Mobilenet pair is a lot. Could you please clarify it, and kindly release the variance of the results reported in the paper, which are important for reproducibility.

Thanks.

---

> ### Author Response · Authors · 2023-11-21
> **Response to Peter Chen**
>
> We appreciate your interest in our paper and your valuable feedback. Below, please find our response to your comment.
>
> > Comment: We are interested in your work. However, after we attempting to reproduce the results in table 2 and table 3, using your provided code, we realized that WTTM is very sensitive to hyper-parameters T and beta, i.e., a change in these two values changes student's accuracy a lot. Also, the variance of student accuracy for the VGG13-Mobilenet pair is a lot. Could you please clarify it, and kindly release the variance of the results reported in the paper, which are important for reproducibility.
>
> **Response**:  For all kinds of distillation methods, high variance is expected on CIFAR-100 dataset, and that’s why it’s a convention to average over multiple runs when reporting results on this dataset.  Moreover, it’s expected that the variance varies from one teacher-student pair to another, owing to the difference among models, and that's why the variance is larger than usual in some cases (e.g. vgg13&#8594;MobileNetV2). Actually, the case is quite different on ImageNet, where the difference among results of multiple runs is basically below 0.1%. Therefore, the high variance of our results is not caused by our method, but the nature of the dataset.
>
> To further alleviate your concern, we revised our paper, providing the standard deviation of our results in appendix A.8 as you requested, together with that of other benchmark methods. It's shown that the standard deviation of our results is comparable to other benchmark methods, verifying that the high variance of results is general on CIFAR-100.
>
> Additionally, to reproduce our results, it's also important to follow our environmental settings specified in the README file of our repository (Python 3.8, PyTorch 2.0.1, and CUDA 11.7), since they could make a nontrivial difference.
>
> Finally, in terms of your concern about hyperparameters, please refer to our response to Reviewer DiN5.

---

### Meta-Review · Area_Chair_e5Xm · 2023-12-06

**Metareview:**

The authors propose a new perspective on the temperature parameter in knowledge distillation. Specifically, they propose to drop the tempering for the student predictive distribution and only apply it to the teacher. The authors connect this proposal to a Renyi divergence regularizer on the student predictions. Empirically, the authors show strong results compared to recent state-of-the-art distillation methods across a range of tasks and models.

## Strengths

The proposal is very simple and easy to implement. The authors perform exhaustive comparisons to multiple recent knowledge distillation baselines and show improved performance. The method leads to practical improvements.

## Weaknesses / Suggestion

Overall, [Stanton et al](https://arxiv.org/abs/2106.05945) shows that the temperature parameter is different depending on the distillation data distribution. Intuitively, the teacher will be over-confident on its own training data. If new data is used for distillation, or if strong data augmentation is used, tempering may not be helpful. I wonder how the proposal from the authors interacts with the distillation data distribution.

**Justification For Why Not Higher Score:**

The reviews are somewhat mixed, and no reviewer strongly championed for the paper. While the results are very encouraging, it is not fully clear still if the method will work as well in new settings, on other distillation tasks, more large-scale models etc.

**Justification For Why Not Lower Score:**

The proposed method appears to be highly practical, outperforming strong recent baselines on a range of tasks. The paper makes a valuable contribution to the distillation literature.

---

### Decision · Program_Chairs · 2024-01-16

Accept (poster)